# Tuberculosis Epididymo-Orchitis Mimicking Malignancy Resulting from Intravesical Bacillus Calmette–Guérin Immunotherapy for Bladder Cancer: A Case Report of a Rare Complication

**DOI:** 10.3390/diagnostics12112663

**Published:** 2022-11-02

**Authors:** Liang-Wei Chiu, Li-Hsien Tsai, Po-Fan Hsieh, Wen-Chi Chen, Chao-Hsiang Chang

**Affiliations:** 1Department of Urology, China Medical University Hospital, Taichung 40447, Taiwan; 2School of Medicine, Graduate Institute of Biomedical Sciences, College of Medicine, China Medical University, Taichung 40402, Taiwan; 3Graduate Institute of Integrated Medicine, College of Chinese Medicine, China Medical University, Taichung 40402, Taiwan

**Keywords:** tuberculosis, epididymo-orchitis, bacillus Calmette–Guérin, bladder cancer, case report

## Abstract

Tuberculous epididymo-orchitis is a rare complication of intravesical bacillus Calmette–Guérin (BCG) immunotherapy for bladder cancer. We report a patient with bladder cancer and a history of intravesical BCG immunotherapy who presented with right scrotal pain for 1 week. A heterogeneous, hypoechoic, and solid mass surrounded by increased blood flow in the right testis was seen on scrotal echogram. Urine ordinary and tuberculosis culture yielded negative results. After failure of antibiotic treatment and the inability to rule out tumor, the diagnosis was confirmed by radical orchiectomy. Acid-fast staining of pus in the tumor and tumor tissue was positive, and a pus culture was positive for *Mycobacteria tuberculosis* complex. Right radical orchiectomy was performed, and anti-tuberculosis treatment with rifampicin, isoniazid, ethambutol, and pyrazinamide was given. The patient is still currently under anti-tuberculosis treatment, and no significant adverse effects have been noted. BCG-related epididymo-orchitis should be suspected in patients with a history of intravesical BCG immunotherapy if the empiric antibiotic treatment typically used to treat common epididymo-orchitis fails.

## 1. Introduction

Bacillus Calmette–Guérin (BCG) is a live attenuated vaccine derived from *Mycobacterium bovis* [1], and it is standard treatment for intermediate- and high-risk patients with non-muscle invasive bladder cancer (NMIBC) after transurethral resection of a bladder tumor (TURBT) [2]. The induction and maintenance of intravesical BCG immunotherapy can significantly decrease the recurrence of bladder cancer. The complications caused by intravesical BCG vary from self-limited irritative voiding symptoms, such as cystitis, epididymo-orchitis, prostatitis, and pyelonephritis, to serious events including hepatitis, pneumonitis, and sepsis [3]. Most patients can tolerate the treatment course smoothly, and only about 5% of patients need to suspend treatment due to severe adverse effects [4]. The early presentation of adverse events (within 3 months after starting instillation) is characterized by generalized non-specific symptoms such as low-grade fever and fatigue. The late presentation of adverse events (after 1 year of treatment) is usually localized to the genitourinary tract and/or other typical sites of mycobacterial infection [3]. The incidence of tuberculous epididymo-orchitis following intravesical BCG is about 0.4% [4]. We report a case of tuberculous orchitis with progressive right scrotal tenderness 1 year after intravesical BCG immunotherapy.

## 2. Case Presentation

A 78-year-old man with hypertension presented with painless gross hematuria for 10 days. A bladder tumor was identified by cystoscopy. He underwent TURBT, and the pathology report showed high-grade papillary urothelial carcinoma without lamina propria or muscle invasion. He received regular follow-up after the first treatment, and recurrence of the bladder cancer was found three years later. After a second operation, he underwent a full course of intravesical BCG immunotherapy (six times weekly), and a maintenance course was scheduled for up to one year. Bladder recurrence with progression to high-grade infiltrating urothelial carcinoma of the bladder with lamina-propria invasion was diagnosed one year later. He refused cystectomy, and therefore received repeated intravesical BCG immunotherapy and pembrolizumab according to the results of KEYNOTE-057 [5].

One year after the latest BCG therapy, he reported progressive right scrotal tenderness. Scrotal ultrasonography showed a heterogeneous, hypoechoic, and solid mass surrounded by an increased blood flow signal in the right testis (Figure 1). According to his symptoms and echogram findings, right epididymo-orchitis was diagnosed, and we prescribed empiric antibiotics with ciprofloxacin 500 mg Q12H for 10 weeks. However, there was no improvement in right scrotal tenderness after antibiotic treatment, and ultrasonography still revealed a solid mass with heterogeneous echogenicity (Figure 2). Testicular cancer screening was then performed, which included lactate dehydrogenase (LDH), alpha-fetoprotein (AFP), and beta-human chorionic gonadotropin (β-HCG), all of which were within normal limits. After discussion with the patient about potential malignancy, he underwent right radical orchiectomy. The specimen showed a 3.0 × 2.5 × 2.0 cm^3^ necrotic tumor containing pus, which had invaded through the tunica albuginea to the tunica vaginalis grossly (Figure 3). Histology revealed necrotizing granulomatous inflammation involving the testis, epididymis, and spermatic cord, with giant cell formation. Acid-fast stain (AFS) of the pus was positive, and a pus culture was positive for *Mycobacteria tuberculosis* (MTB). AFS of tumor tissue was positive, and MTB quantitative polymerase chain reaction (PCR) from the pus was also positive. The diagnosis of post-BCG tubercular orchitis was confirmed. He was then referred to the infectious outpatient department, where he received treatment with rifampicin, isoniazid, ethambutol, and pyrazinamide for nine months.

## 3. Discussion

The incidence of BCG-related epididymo-orchitis is rare, accounting for 0.4% of all BCG-related complications [4]. The onset of BCG-related orchitis ranges from 2 weeks to 17 years [6,7], and our patient presented one year after BCG instillation. He did not have diabetes. Dooley et al. reviewed the relationship between diabetes and tuberculosis and reported that there was a high risk of developing complications from BCG instillation in diabetic patients [8]. However, a previous or a latent tuberculosis infection does not seem to increase the risk of BCG complications [9]. For studying whether prophylactic use of ofloxacin is effective in the protection against BCG complications, Colombel et al. conducted a randomized, prospective, and double-blind placebo-controlled trial [10]. The results indicated that the participants receiving 200 mg ofloxacin in the study group had an 18.5% decreased risk of BCG-related adverse events compared to the control group.

Our patient also received pembrolizumab according to the result of KEYNOTE-057 [5]. It combined with repeated intravesical BCG immunotherapy. Alanee et al. reported that the intravesical BCG combined with intravenous pembrolizumab therapy was safe, with grade 3/4 adverse effects representing 11% of overall events [11]. However, whether BCG-pembrolizumab combined therapy increased the risk of tuberculous epididymo-orchitis was still unclear. Further observation of disease progress in this patient is warranted.

It can be difficult to differentiate tuberculous epididymo-orchitis from other common bacterial infections clinically. The common presentations of tuberculous epididymo-orchitis include non-specific symptoms such as dysuria, scrotal swelling, and pain. Scrotal ultrasound may be useful for the diagnosis.

The ultrasound features of BCG-related epididymo-orchitis include a heterogeneous, hypoechoic, and solid mass with decreased flow, but surrounded by hyperemia. It corresponds to granuloma formation and can be used to differentiate BCG-related epididymo-orchitis from non-mycobacterial infection [12]. Other ultrasound features include diffuse hypoechoic heterogeneous enlargement, diffuse hypoechoic homogeneous enlargement, and multiple small hypoechoic nodules in the testicles [7]. It may mimic malignancy, and if the mass extends beyond the tunica albuginea, malignancy should be suspected [13].

The diagnosis of tuberculous epididymo-orchitis is made if there is a typical clinical presentation combined with one of the following criteria: (1) positive AFS in urine; (2) positive urine culture for *M. tuberculosis*; (3) positivity of PCR for *M. tuberculosis* in urine; and (4) typical granulomatous inflammation with caseous necrosis in microscopy plus any positive result of Ziehl–Neelsen AFS or PCR for *M. tuberculosis* in any relevant tissue specimen [14,15]. However, because of the lack of diagnostic methods with high sensitivity and specificity, tuberculous epididymo-orchitis is often misdiagnosed as a bacterial infection or tumor [14,16].

From a clinical point of view, tuberculous epididymo-orchitis should be suspected in patients with a history of intravesical BCG immunotherapy, and further evaluation including urine-base tuberculosis screening should be prescribed initially. Premature empiric antibiotic treatment might cover the signs of infection, which might make a delay in diagnosis. In our patient, the initial symptoms were right scrotal pain and swelling, and scrotal ultrasound findings included a heterogeneous mass with increased blood flow in the right testis. The patient’s symptom did not improve after 10 weeks of empiric antibiotics, and follow-up ultrasound revealed a progressive granuloma-like lesion with peripheral hyperemia. The image findings raised the suspicion of malignant change; however, testicular tumor markers did not provide enough evidence to rule out malignancy. Radical orchiectomy was then arranged as a diagnostic and therapeutic procedure, and the pathology report showed tuberculous epididymo-orchitis with positive AFS.

In patients with a history of intravesical BCG and ultrasonography features of testis lesions, BCG-related tuberculosis orchitis should be highly suspected. Urine-based tuberculosis screening and magnetic resonance imaging (MRI) before surgery may provide some clues to avoid misdiagnosis [17].

## 4. Conclusions

Intravesical BCG immunotherapy is a useful treatment for NMIBC after TURBT; however, BCG-related tuberculosis epididymo-orchitis should be kept in mind. Although the symptoms and signs of tuberculosis epididymo-orchitis are often nonspecific, ultrasound and urine-based tuberculosis screening before empiric antibiotic treatment can be used to make the diagnosis. It can be difficult to differentiate BCG-related epididymo-orchitis from malignancy based on the clinical presentation and image findings. Orchiectomy can be performed as a diagnostic and therapeutic procedure.

## Figures and Tables

**Figure 1 diagnostics-12-02663-f001:**
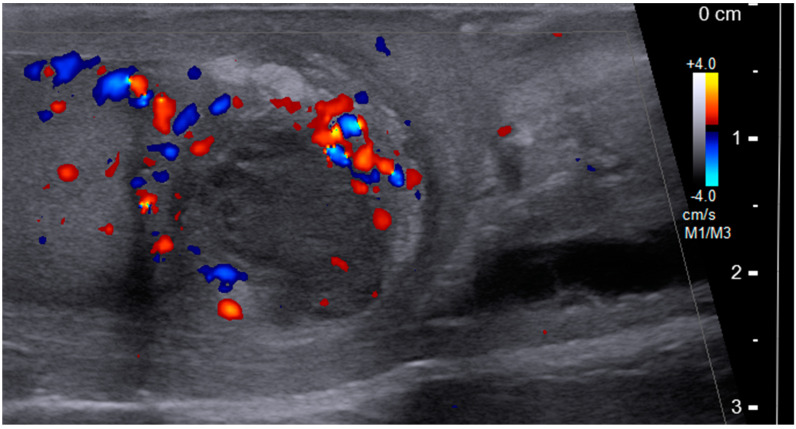
Scrotum ultrasound showing a heterogeneous, hypoechoic, and solid mass with increased blood flow.

**Figure 2 diagnostics-12-02663-f002:**
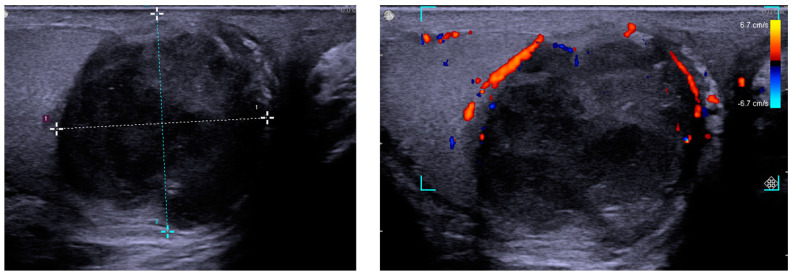
Scrotum ultrasound showing a 28 × 29 mm^2^ heterogeneous, hypoechoic, and solid mass with decreased blood flow, but surrounded with increased blood flow.

**Figure 3 diagnostics-12-02663-f003:**
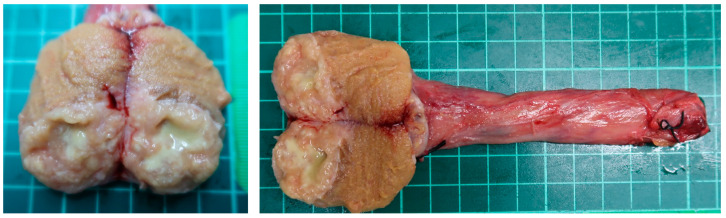
Specimen showing a 3.0 × 2.5 × 2.0 cm^3^ necrotic tumor with pus content in the testes, which had invaded through the tunica albuginea to the tunica vaginalis grossly.

## Data Availability

All of the data are available upon request to the corresponding author.

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
