# Peer review of "Tuberculosis Epididymo-Orchitis Mimicking Malignancy Resulting from Intravesical Bacillus Calmette–Guérin Immunotherapy for Bladder Cancer: A Case Report of a Rare Complication"

_diagnostics, 2022, doi:10.3390/diagnostics12112663_

Round 1

Reviewer 1 Report

Complications following intravesical BCG treatment should be known and it is important to realize that these complications can occur long time following BCG instillations.

Initially, the patient was treated with antibiotics although there were no signs of infection especially with orchitis the patient should have fever, therefore, a delay in diagnosis was made. 

Thus. this should be changed in the recommendations, there should not be a empiric AB treatment.

Reviewer 2 Report

Liang Wei-Chiu et al. described a case of tuberculosis epididymo-orchitis mimicking tumor as an effect of intravesical BCG therapy for BCa.

Generally the manuscript is well organized and written, following the rules for case report.

However, its substantive value in the category of a scientific report is doubtful and, in my opinion, it does not bring anything innovative. Testicular tumors quite often mimic the symptoms of epididymitis. On the other hand, testicular tumor rarely occurs in elderly patients, and the fact that epididymitis may be one of the symptoms of urogenital tuberculosis is well known.

An interesting fact in this case is the combined therapy - BCG + checkpoint inhibitor - analysis of this issue could influence the "attractiveness and specificity" of the presented case. This thread is not mentioned by the authors in the article - especially in the discussion.

Round 2

Reviewer 2 Report

Authors adressed comments.